# Barriers and promoters to adapting research findings to clinical care in hereditary angioedema in the United States: A qualitative study

Esther L. Langmack[1]*, Dana Ravyn[2], Rob Lowney[2], Beth Goodwin[2], William R. Lumry[3]

1 Langmack Medical Communications, Denver, Colorado, United States of America, 2 CMEology, West Hartford, Connecticut, United States of America, 3 Department of Internal Medicine, Allergy and Immunology Division, University of Texas Southwestern Medical School, Dallas, Texas, United States of America

☯ These authors contributed equally to this work.
* elangmack@gmail.com

## Abstract

Treatment options for hereditary angioedema (HAE) have evolved due to discoveries in basic and clinical research. HAE clinical guidelines emphasize optimizing quality of life through attack prevention strategies that include long-term prophylaxis. We sought to identify barriers to and promoters of research translation into HAE clinical practice that may inform efforts to improve patient care. We interviewed US allergy/immunology clinicians who completed an online continuing medical education activity on HAE or were identified through healthcare provider directories. Interviews focused on clinicians' experiences translating HAE research results into clinical care. Deidentified interview transcripts were coded and analyzed to detect emergent themes using Dedoose software. Thematic analysis of 15 interviews showed that insurance prior authorization for HAE medications, including those for long-term prophylaxis, was perceived as the biggest barrier to evidence-based care. Other prominent barriers were laboratory testing difficulties, time constraints, and deficits in both primary care provider and patient understanding of HAE. Promoters of research translation included availability of medication samples, route/frequency of drug administration, and shared decision-making. Clinicians used different resources to learn about HAE, including online chat rooms. None of the clinicians used a validated instrument to assess HAE-related quality of life. Perceptions of the usefulness of HAE clinical guidelines were mixed. A range of factors act as barriers and promoters to research translation into clinical care for patients with HAE. Our findings have implications for interventions to enhance the delivery of evidence-based care for patients with HAE.

**Data availability statement:** All data and code files for this study are available from the Harvard Dataverse database at https://doi.org/10.7910/DVN/ZBIFEF.

**Funding:** This research was supported by an independent educational grant from Takeda Pharmaceuticals USA, Inc. (Grant IME-007727) to CMEology, West Hartford, CT. The funders of this study had no role in study design, collection, analysis, interpretation of data, writing the report, or the decision to submit the report for publication.

**Competing interests:** I have read the journal's policy and the authors of this manuscript have the following competing interests. Esther L. Langmack, Dana Ravyn, and Beth Goodwin state they have no competing interests to disclose. Rob Lowney discloses that this research was supported by an independent educational grant from Takeda Pharmaceuticals USA, Inc. (Grant IME-007727) to CMEology, West Hartford, CT. William R. Lumry discloses the following: grants or contracts (Astria, BioMarin, CSL Behring, Intellia, Grifols, Ionis, Kalvista, Shire/Takeda); consulting fees (Astria, BioCryst, Biomarin, CSL Behring, Express Scripts/CVS, Fresenius Kabi, Intellia, Kalvista, Magellan, Optum, Pharming, Pharvaris, Shire/Takeda); payments or honoraria for lectures, etc. (BioCryst, CSL Behring, Optinose, Pharming, Shire/Takeda, Grifols, Astra Zeneca, Sanofi/Regeneron, GSK); payment for expert testimony (Vedder-Price, Murphy & King); leadership for fiduciary roles (US Hereditary Angioedema Association Medical Advisory Board). This does not alter our adherence to PLOS ONE policies on sharing data and materials.

## Introduction

Hereditary angioedema (HAE) is characterized by recurrent and often unpredictable episodes of submucosal and cutaneous swelling without urticaria that typically last for several days [1]. Angioedema may affect the extremities, intestinal tract, or other organs, causing pain and disfigurement and interfering with work, school, and social activities. HAE imposes a significant psychological burden and reduces quality of life (QoL) [2,3]. Attacks involving the upper airway or larynx can cause fatal asphyxiation. Approximately 50% of patients experience a laryngeal attack at some point in their lives, but there is no reliable way to predict who will experience a laryngeal attack. With an estimated prevalence of 1 in 50,000 persons, HAE is a rare disease that often goes undiagnosed or misdiagnosed for years [1].

Pharmacologic management of HAE consists of acute treatment of angioedema attacks, short-term prophylaxis (e.g., for dental procedures), and long-term prophylaxis. The advent of new disease-specific prophylactic therapies for HAE that are safe and effective and that patients can administer themselves at home has made it possible for many patients to live normal lives [4,5]. The most recent evidence-based management guidelines for HAE state that maximizing patient QoL and preventing all attacks are the overarching treatment goals for patients with HAE [4,5].

The purpose of this qualitative study was to identify barriers to and facilitators of research translation into clinical care for patients with HAE. Little is known about research translation in HAE, especially from the viewpoint of allergy/immunology clinicians. Translational science is the study of how observations or evidence from biological, public health, or clinical research become interventions aimed at enhancing human health [6]. Qualitative research methods are widely used in translational science to elicit the perspectives of stakeholders, including clinicians, and to understand factors that discourage or facilitate implementation of scientific advances in clinical practice [7]. By identifying factors that are pivotal in research translation, interventions or practice changes may be discovered that could enhance delivery of care to patients with HAE.

## Methods

The continuing medical education (CME) activity *Case Studies in Hereditary Angioedema: Moving Beyond Crisis to Long-Term Prevention* was available for 12 months on the educational platform myCME.com starting on August 2, 2023. The activity was intended for allergists, immunologists, primary care providers (PCPs), and advanced practice providers involved in those specialties. The activity was certified for 1.5 *AMA PRA Category 1 Credits*™ by Partners for Advancing Clinical Education (PACE). The faculty person for the activity was one of the authors (WRL).

The study was verified to be exempt according to 45 CFR 46.104(d) (2) by an independent review board (Solutions IRB) on December 13, 2023. The recruitment process for this study started on January 9, 2024, and ended on June 27, 2024. Beginning on January 9, 2024, US learners who had completed the CME activity were asked by email if they wanted to be interviewed about the translation of research results into patient care related to HAE. Learners who were interested in

participating and were allergy/immunology clinicians were invited by email to be interviewed. Additional allergy/immunology physicians were invited from a geographically diverse range across the United States, as identified through healthcare provider directories.

The risks and benefits of participation in this study were explained and presented in writing to all potential interviewees prior to the interview. All participants provided verbal consent before starting the interview. One investigator (ELL; physician) obtained and documented the verbal consent and conducted the interview online (audio only). Audio recordings were transcribed verbatim. Participant data and personal information were deidentified. Data were read, analyzed, and coded by one investigator (DR) with coding reviewed by all investigators. Interviews and notes were coded using Dedoose, a web-based application for qualitative analysis and mixed methods research [8]. Data and codes used in this study are available at https://doi.org/10.7910/DVN/ZBIFEF [9].

The qualitative study was designed to enhance understanding of research translation in the allergy/immunology clinical setting through thematic analysis. The study used grounded theory methodology [10] and employed a hybrid deductive and inductive analysis [11,12]. Deductive analysis used pre-empirical sensitizing concepts and codes from existing research, while the inductive analysis used post-empirical concepts and codes derived from the examination of the interview data [13].

Dynamic semi-structured interviews using open-ended questions were conducted to gather data about clinician knowledge, attitudes, opinions, beliefs, feelings, and practices. These observations were used to characterize the relationship between existing and conceptual frameworks of the clinicians' beliefs and practices related to translation of evidence to patient care while identifying significant emerging concepts and themes.

We designed the interview guide and obtained IRB approval before the research began, although participants were encouraged to offer additional views that were relevant and within the framework of the study protocol. Thematic analysis was initially performed by DR, then reviewed and revised by all authors.

A pre-determined target of 15 interviews was set based on previous studies demonstrating that a minimum of 6–12 interviews would be needed to reach a state of saturation, defined as the point at which reviewing additional data yields no new themes in thematic analysis [14,15]. We assessed saturation using the validated methods of Guest et al. [16]. This approach measures the *saturation ratio*, a fraction wherein the denominator is the *base size* (the minimum number of data collection events to be analyzed in order to calculate the amount of information already accumulated), and the numerator is the *run length* (the number of interviews within which new information is sought). The *new information threshold* is the maximum size of the resulting proportion accepted as evidence that saturation has been achieved at a given point in the dataset [16].

## Results

A total of 2184 healthcare professional learners participated in the online CME activity. Learners had a mean of 12.5 years in practice and reported seeing an average of 5 patients with HAE annually. Confidence levels were assessed on a 5-point Likert scale (1 = not very confident to 5 = very confident). Among allergy/immunology clinicians, self-reported confidence in the clinical and laboratory diagnosis of HAE and the use of long-term prophylaxis to decrease disease burden and improve QoL increased following the activity (HAE diagnosis: pre-activity [mean, SD], 3.89 [1.94] vs post-activity, 4.37 [1.89]; Cohen's d = 0.251. Long-term prophylaxis: pre-activity, 3.68 [2.09] vs post-activity, 4.32 [2.23]; Cohen's d = 0.296). Allergy/immunology clinician knowledge of different elements of HAE diagnosis and treatment also increased after the activity (CMEology, data on file).

Nine learners (6 physicians, 2 physician assistants [PAs], 1 nurse practitioner [NP]) from the CME activity and 6 physicians identified through healthcare provider directories volunteered to be interviewed (Table 1). Physicians identified through provider directories were comparable to the overall CME activity physician participant population in terms of experience, practice setting, age, and gender. A total of 15 individuals, all of whom were allergy/immunology health

**Table 1. Participant characteristics.**

| Participant | Gender | Age (decade) | Degree | Years in practice | Practice setting | US region | Recruitment method |
|---|---|---|---|---|---|---|---|
| 1 | M | 20s | PA | 3 | Group | S | CME activity |
| 2 | F | 20s | PA | 6 | Group | S | CME activity |
| 3 | F | 30s | NP | 9 | Group | S | CME activity |
| 4 | F | 30s | MD | 7 | Group | NE | CME activity |
| 5 | M | 30s | MD | 11 | Group | NE | CME activity |
| 6 | F | 30s | MD | 11 | Group | NE | Directory |
| 7 | M | 30s | MD | (missing) | Group | W | Directory |
| 8 | F | 40s | MD | 14 | Group | NE | Directory |
| 9 | F | 40s | MD | 14 | Group | S | Directory |
| 10 | M | 40s | MD | 19 | Group | W | CME activity |
| 11 | M | 40s | MD | 20 | Group | S | Directory |
| 12 | M | 40s | MD | 20 | Group | NE | Directory |
| 13 | M | 50s | MD | 25 | Group | S | CME activity |
| 14 | M | 50s | MD | 25 | Group | NE | CME activity |
| 15 | M | 60s | MD | 35 | University | NE | CME activity |

CME, continuing medical education; MD, medical doctor; NE, Northeast; NP, nurse practitioner; PA, physician assistant; S, South; W, West.

professionals practicing in the United States, completed an interview lasting 37.3 minutes, on average, between January 16, 2024, and June 27, 2024.

We used a saturation ratio calculation [16] and found that we reached saturation at new information thresholds of ≤0.05, ≤0.02, and zero with 10, 11, and 13 interviews, respectively. We concluded that 15 interviews was a sufficient number.

Thematic analysis of the interview data revealed that participants had varying levels of experience and success in evaluating the HAE research and translating research findings into clinical practice. However, several prominent and recurrent themes emerged regarding barriers and promoters of research translation. Representative quotes from participants related to main themes, concepts, and factors are included below.

### Barriers to research translation

Three main themes regarding obstacles to research translation in HAE care were identified: institutional barriers, practice barriers, and patient barriers (Table 2). Insurance prior authorization was by far the most frequently cited barrier in any of the 3 main barrier themes.

### Institutional barriers

Institutional barriers included the insurance prior authorization process and issues related to laboratory testing for HAE. Prior authorization for HAE medications from insurers places significant demands on clinicians and office staff. Some participants perceived that insurance companies were not sufficiently familiar with the current HAE clinical guidelines, making it even more difficult to obtain approval for HAE medications. Due to frustrations with prior authorization, some individuals felt discouraged from prescribing the latest treatments described in the literature and avoided the process altogether. In some cases, an insurer declined to authorize a medication unless the patient had already tried it, yet the clinician did not have access to medication samples for the patient to try.

*The insurance companies sometimes will create barriers that don't go along with guidelines or things that have been recommended by thought leaders or places that are seeing [more] patients… or like the clinical history or laboratory*

**Table 2. Barriers and promoters of research translation into clinical practice in HAE.**

| | Theme | Concept | Factor (count)[a] |
|---|---|---|---|
| **Barriers** | **Institutional** | Institutional barriers that prevent clinicians from initiating evidence-based care | • Prior authorization (19)<br>• Testing[b] (7) |
| | **Practice** | Clinician practices that result in suboptimal translation of research to care | • PCP lack of knowledge or misdiagnosis (9)<br>• Lack of information (8)<br>• Clinician time constraints (5)<br>• Clinical inertia (4)<br>• Cases too complex for primary care (2)<br>• Many HAE treatments to choose from (2)<br>• Lack of resources to facilitate communication about HAE with patients (1) |
| | **Patient** | Patient issues that result in interruptions or inability to initiate evidence-based care | • Patient time constraints (6)<br>• Patient anxieties/concerns (5)<br>• Patient lack of knowledge about HAE (3)<br>• Long-term safety/pregnancy concerns (2)<br>• No show/adherence (2) |
| **Promoters** | **Institutional, Practice, or Patient** | Clinical characteristics that promote research translation | • Availability of medication samples (7)<br>• Patient-centered decision-making/shared decision-making (4)<br>• Route/frequency of medication administration (3)<br>• Fear of HAE attacks (2)<br>• Improved clinician knowledge or skills (1) |
| | **Clinician strategies** | Clinicians rely on specific resources to facilitate efficient research translation | • Literature review (18)<br>• UpToDate/Google/others (14)<br>• CME (14)<br>• Colleagues (14)<br>• Guidelines (11)<br>• Professional conferences (posters, presentations) (10)<br>• Pharmaceutical representative detailing (8)<br>• Professional society updates/tests (4)<br>• Online chat rooms/forums (2)<br>• Specialty pharmacist (1)<br>• Press releases (1) |

CME, continuing medical education; HAE, hereditary angioedema; PCP, primary care provider.

[a]The number of times that each factor occurred in interviews. A factor could have been mentioned more than once by a single participant.

[b]Includes lack of access, no definitive test (Type III HAE), delays, intra-laboratory variation, loss of patient to follow-up.

*findings, they make these hoops to jump through that you have to end up doing peer reviews and so forth, which can be quite frustrating because a lot of times if you do a peer review, it's not someone of our specialty. (MD, group practice)*

*… the big hassle comes about when I have to pull resources. I have dedicated employees that have to come in, so they're getting an hourly salary, and then they have to get these things prior authorized. (MD, group practice)*

*In essence, you can't get anything approved without trying [it in the patient first], but you won't be able to try because the institution doesn't want you to have samples in the office. So, both of those are huge; it just makes things very difficult. (MD, group practice)*

*… we've had such a hard time on some of these PA's [prior authorizations] that typically, I'm sort of limiting [what I prescribe] to what I think we can simply get. (MD, group practice)*

Another institutional barrier was trouble obtaining specific laboratory tests or ensuring the accuracy or reliability of tests (e.g., C1 esterase inhibitor [C1-INH], complement, or genetic tests). Delays in receiving test results and loss of patients to follow-up were related challenges. These barriers made it more difficult for participants to diagnose HAE.

*Maybe it's difficult to obtain the lab test to even diagnose the patients first off; maybe they don't offer the test; maybe they're send-outs and that takes a lot of time to [get a] result. (MD, group practice)*

*A lot of the testing is difficult to do. It's not so simple to send genetic testing on these patients. And then, a lot of times, you have a value, you want to repeat the value, it's hard to get the patient to come back. It's an intermittent disease, and that creates a very big issue. (MD, group practice)*

**Practice barriers**

Practice barriers consisted of clinician knowledge, attitudes, and behaviors that resulted in suboptimal translation of research to patient care. Clinicians lacked sufficient time to keep up with the latest developments in HAE. The number of articles or conference presentations on HAE is much smaller than it is for a common condition such as asthma. Interviewees used a variety of sources to locate information about HAE, but many found it challenging to have the right information at the right time to support clinical decision-making. Information sources included online practice summaries (e.g., UpToDate [17]), Google searches, information supplied by pharmaceutical companies or professional societies, journals, CME activities, and presentations at national or regional allergy/immunology meetings.

*In the office, time is always a limitation … for me, that's definitely a barrier. I don't even have a nurse right now, so I'm giving my own allergy shots, my own food challenges, so it's a barrier. (MD, group practice)*

*… we do use the internet … we don't always have the information ready at hand just because we don't come across them [patients with HAE] a lot … we don't have a lot of pharmaceutical [representatives] coming to us to talk. (PA, group practice)*

Participants also identified deficits in awareness, knowledge, competence, and performance among PCPs and other health professionals as impediments to optimal care.

*… a lot of times, we have to educate the PCP about what it is … providers don't consider it and aren't aware of it. They thought she had like a tuna fish allergy. It was a hospital consult. I mean, we see a lot of times giving epi [epinephrine], giving steroids, thinking it's an allergic reaction. (MD, group practice)*

Regarding pharmacologic therapy for HAE, participants stated that hesitation to prescribe new treatments and the increasing complexity of the HAE treatment landscape have hindered research translation into practice.

*I think some providers might see it, you know, if they've been prescribing certain medications that have been working, they might be kind of like, well, "Why start doing something different?" (NP, group practice)*

*… as biologics and other specialty drugs move into allergy/immunology practice, it's not as simple as it used to be. (MD, group practice)*

**Patient barriers**

Patient-related factors resulted in interruptions to care and an inability to deliver evidence-based care. Participants described patients' unmet need for education about HAE and the challenges of engaging them in treatment decisions and ensuring proper follow-up.

*… a lot of times, we'll say that the patient doesn't really fully grasp it as far as the medical side of it … [getting them to an] understanding of what exactly is going on, so we can try to simplify the mechanism of action or the plan of care for the next 6 months. (PA, group practice)*

*I think patients are a little bit more accepting, but still, they all want to know long-term safety. (MD, group practice)*

*I think there is a lot of noncompliance generally with HAE, especially if people are on injectables because people don't like injectables. (MD, group practice)*

*I've had patients that just aren't very compliant with visits; it's hard to get them to come in to kind of see what's going on. And then, because they're not coming in for a visit, then insurance won't pay for the next set of medications and things like that. (MD, group practice)*

## Barriers related to long-term prophylaxis for HAE

Interviewees were asked about their experience evaluating patients for long-term prophylaxis and how they assessed patient QoL. Overall, barriers to long-term prophylaxis mirrored those identified for HAE care and included struggles with prior authorization and lack of patient knowledge about HAE. Other patient-related barriers included hesitancy to change their medication regimen and the difficulty of weighing attack prevention against the commitment of time and resources associated with starting a new, regularly scheduled medication (Table 3).

*… having them truly understand that this is somewhat of a chronic disease … having discussion between rescue therapy versus maintenance therapy is a little bit difficult with some of these patients. (MD, group practice)*

*Some patients just really don't like medications; they don't want to be taking something so frequently, even if it is preventative. Some people, they just prefer like,"You know what, if I get an attack, I'll just treat it." (MD, group practice)*

*So long-term prophylaxis, I think anytime you're talking about something that's forever, it's going to be a difficult discussion for people. (MD, group practice)*

*… someone who has mild issues with episodic [attacks], then they're like,"Well … I can't afford it." (MD, group practice)*

Clinician uncertainty about whether a patient was a suitable candidate for prophylaxis and hesitancy to try a new medication were also barriers to initiating long-term prophylaxis.

**Table 3. Barriers and promoters of research translation specific to long-term prophylaxis for HAE[a,b].**

| Barriers | **Physician-related**<br>• QoL self-assessment instruments not used (16)<br>• Hesitance with new products (2)<br>**Patient-related**<br>• Patient lack of knowledge about HAE (9)<br>• Patient time constraints (6)<br>• Few episodes/prefers episodic treatment (5)<br>• Patient anxiety/concerns (5)<br>• Cost of medication (3)<br>• Pharmacy (1) |
|---|---|
| Promoters | • History of more severe/frequent HAE attacks (3)<br>• Alleviating fear of serious attack (2)<br>• Impact on QoL (2)<br>• Efficacy and safety data (1) |

HAE, hereditary angioedema; QoL, quality of life.

[a]The number of times that each barrier or promoter occurred in interviews is shown in parentheses. A barrier or promoter could have been mentioned more than once by a single participant.

[b]Other barriers and promoters of research translation discussed above may also apply.

*… you get a random positive [test result for HAE] and you don't know what to make of it; and if they are symptomatic... it becomes a question of, do these people need preventative therapy or do we just have rescue for them? (MD, group practice)*

*… you never want to be the first kid on the block to use something. (MD, group practice)*

*I think anytime that new things come out, there is a little bit of hesitancy or fear just with the unknown, so it's kind of just about jumping in and trying something different, especially if it's going to improve the quality of life of patients. (NP, group practice)*

### Promoters of research translation

Factors that were viewed as promoting research translation into HAE care, in general, included availability of medication samples, route of medication administration (e.g., patient preference for oral vs subcutaneous route), and use of patient-centered decision-making (Table 2).

*Without an ability to get a [medication] sample, it would be really difficult to treat these patients. (MD, group practice)*

*I think lately, I've been recommending the newer options just because they're easier, they're very effective, easier routes of administration, things like that. (MD, group practice)*

*I think some people are okay with taking a pill every day; some people are not. Some people prefer the injection. So, I think it has to be in the discussion. But for me, it's what they're going to be most compliant with. (MD, group practice)*

### Promoters related to long-term prophylaxis for HAE

Several factors promoted consideration of long-term prophylaxis by patients and clinicians (Table 3). Attack severity and frequency were the most common factors; less attention was given to QoL and safety and efficacy data. When asked specifically how they assessed the impact of HAE on patients' lives, none of the interviewees reported using a standardized instrument to assess QoL. Some participants believed that an open interview was sufficiently rigorous, while others were either unaware of QoL instruments for HAE or perceived them to be useful only in research settings.

*Well, all of my patients, I have on long-term prophylaxis, so 100% of my patients are on both long-term prophylaxis and acute on-demand therapy. It's always been my approach, and I probably am more conservative than others, but I don't want to put them at risk for having a potentially life-threatening laryngeal attack. (MD, group practice)*

*But sometimes the number [of attacks] is relatively low, but then it could be the severity of the attack, like what organ was involved. And then, some of it is also the perception of the patient that you pick up on, like how, I guess for lack of a better term, traumatized they were by that event. (MD, group practice)*

*I think just considering the total effect on the patient, the condition, just remembering, like I said, the psychological aspects of it and the social issues that have to be considered. (MD, group practice)*

*I am definitely bringing up prophylaxis … recommending it to these patients to improve their quality of life. (MD, group practice)*

### Strategies to enhance research translation

Participants relied on a variety of strategies to facilitate efficient translation of scientific evidence into HAE practice. While participants often used online information sources, interactions with peers and colleagues were valuable opportunities to

learn how other clinicians manage HAE. Two younger clinicians used online chat rooms or forums to discuss HAE diagnosis and management with allergy/immunology clinician peers. In practices where pharmaceutical detailing was allowed, in-person interactions with company representatives were described as useful for obtaining medication samples and streamlining the prior authorization process.

> *I also feel like in this era, it's very easy to find everything on the computer and do CME that way and to keep up that way. (MD, group practice)*

> *I definitely chat [online] with my colleagues, my physician colleagues, not just within my practice but friends of mine, allergy friends. So, I think if I have a clinical question, management question, then I certainly utilize that. (MD, group practice)*

> *I'm in a bunch of different allergy groups both online and then on like WhatsApp and through texts with various colleagues. (MD, group practice)*

> *… we do have a lot of the [pharmaceutical company representatives] come in, they give us samples, they really walk us through if we want to apply for one of the medications for our patients, like get it approved. (MD, group practice)*

Awareness and use of HAE clinical guidelines was mixed, and some reported using them rarely or not at all. Apart from their application in clinical care, guidelines were used to advocate for coverage of an HAE medication during the prior authorization process.

> *I'm not actively reviewing guidelines or treatment decisions. I think as far as the literature, I don't really review anything on a regular basis. I don't feel like I've made many decisions based on any of the clinical practice guidelines. (MD, group practice)*

> *I still try to follow evidence-based medicine and try to follow standards of care, which I think are often established by clinical guidelines that seem to come from like reputable sources. And so, I do try to follow them as much as I can within reason, but as the name says, they're guidelines; they're not rules or mandates, I suppose. (MD, group practice)*

> *Clinical guidelines are good because then the insurance companies can't deny things if it's a respectable institution that develops these guidelines or endorses [them]. (MD, group practice)*

## Discussion

In this qualitative study, analysis of interviews with allergy/immunology clinicians revealed prominent barriers to and promoters of research translation into the care of patients with HAE.

Difficulties with prior authorization and laboratory testing were institutional barriers that interfered with efforts to initiate evidence-based care. The greatest barrier to application of evidence by far was the prior authorization process. Clinician barriers included constraints on their time, lack of information about HAE management, clinical inertia, and limitations in PCP awareness and knowledge of HAE. Patient barriers included lack of knowledge about HAE and concerns about treatment, in addition to difficulties with clinic attendance and medication adherence.

Promoters of research translation were the availability of medication samples to facilitate a treatment trial pending insurance authorization, route and frequency of drug administration, patient-centered or shared decision-making, and improved clinician knowledge and skills related to HAE. Staying current on HAE was difficult, largely because HAE is a relatively rare condition. Compared with diseases such as asthma or allergic rhinitis, clinicians encounter few patients with HAE, and they are exposed to less information about HAE through publications, newsletters, and conferences.

 

Participants relied on a wide variety of information resources to facilitate translation of evidence into practice, including online and in-person sources. The use of online chat rooms or forums in which some younger participants sought information about HAE from peers was a novel finding of this study that may have implications for care in low-resource settings (e.g., rural areas). Further study is needed to show how clinicians use online interactive communication platforms in lieu of more traditional "curbside consult" methods.

Awareness and use of HAE clinical practice guidelines was decidedly mixed in this study, which is consistent with previous implementation research on guidelines. Research has shown that guidelines generally have limited impact on physician behavior, due to a variety of barriers [18]. Translation of new findings into clinical practice can take years or even decades, during which significant opportunities to optimize patient outcomes can be lost [19].

One of the goals of this study was to examine barriers and facilitators of research translation related to the use of long-term prophylaxis, an expanding area of HAE therapeutics. Many barriers and facilitators of HAE care, in general, also applied to long-term prophylaxis, including challenges with prior authorization. Long-term prophylaxis also presented unique challenges, such as (1) the perception that it is not needed if episodes are infrequent or less severe, (2) clinician and patient hesitance to commit to prophylaxis, (3) medication cost, and (4) the need to educate patients about HAE and treatment options. HAE clinical guidelines and experts now recommend individualized assessment of QoL and disease burden in decisions about long-term prophylaxis, with the goal of normalizing patients' lives as much as possible [2,4,5]. Previous HAE guidelines primarily emphasized assessment and reduction of HAE attack frequency and severity, not improving QoL [2]. Very few participants were aware of validated instruments to assess QoL of patients with HAE, and none reported using such tools in their practice. If they were aware of QoL instruments for HAE, they perceived them as impractical or useful only in a research setting. Participants instead used informal interview strategies to assess QoL, which yielded information of varying depth and specificity. The optimal use of validated HAE QoL and disease control questionnaires in clinical practice remains an area of ongoing investigation [2]. However, such tools could potentially help make shared decision-making discussions between patients and clinicians more focused and efficient.

The results of this study build and expand upon the findings of quantitative survey studies of allergy/immunology health professionals [20,21]. One study described challenges related to misdiagnosis of HAE, misinterpretation of test results, and other problems with laboratory testing [20]. Additional challenges were lack of clinician and patient knowledge of HAE, patient inability to afford or access treatment, and difficulties incorporating patient preferences into treatment decisions [20]. Regarding long-term HAE prophylaxis, physicians and patients were previously shown to have differing views of treatment burden [21]. Physicians were more likely than patients themselves to view prophylactic medications administered subcutaneously or intravenously as burdensome, inconvenient, or unpleasant. These findings and those of our study highlight the need for enhanced patient-physician communication and shared decision-making about long-term prophylaxis options.

Difficulty obtaining insurance prior authorization for HAE medications, including those used for long-term prophylaxis, was the most prominent barrier to research translation described by participants in our study. Prior authorization is a common procedure used by medical insurance companies in the United States to ensure that only patients meeting pre-determined diagnostic and other criteria receive financial coverage for selected medications [22]. A qualitative study employing interviews with patients with HAE found that insurance delays, denials, and coverage changes were associated with an increased need for urgent or emergency medical care, more missed days of work or school, heightened patient anxiety levels, and a negative effect on patients' families [22]. Clinicians interviewed in our study described their dissatisfaction with the burdens that the prior authorization process placed on their time and their staff's time.

One strength of this study is that we uncovered several factors not previously reported in quantitative survey studies of allergy/immunology clinicians that may impact delivery of high-quality HAE patient care [20,21]. Specifically, we detected clinician difficulties with accessing information on HAE, unfamiliarity with HAE clinical practice guidelines, and lack of use of QoL assessment tools as barriers to research translation. Provision of medication samples and pharmaceutical

representative detailing were identified as facilitators of care delivery in our study but not previous quantitative survey studies [20,21]. In translational health research, qualitative methods are increasingly recognized as valuable ways to reveal stakeholder attitudes, perceptions, and experiences that escape detection using quantitative approaches [23]. Additionally, this study is the first, to our knowledge, to use contemporary qualitative research methods including computer-assisted thematic analysis to examine the experiences of allergy/immunology clinicians related to HAE patient care. Recent qualitative research on HAE has focused on patients rather than clinicians [3,22,24,25]. However, the experiences and perspectives of allergy/immunology clinicians, particularly those in general allergy/immunology practices (where most of our respondents worked), also need to be more fully recognized if treatment is to improve for all HAE patients, not just those who receive care in HAE subspecialty centers.

Potential limitations to this study include the fact that it enrolled a non-random sample of US allergy/immunology health professionals, the majority of whom were general allergy practitioners in group practices, whose perspectives may potentially differ from those of a broader group of providers. Barriers and promoters relevant to research translation into HAE clinical practice may be different outside of the United States, for example in Europe, where many patients with HAE receive care in rare disease centers from HAE specialists rather than general allergy practitioners. Additionally, insurance prior authorization would not be a barrier in countries in which the public healthcare system does not require this type of review. Although clinicians in our study reported seeing an average of only 5 patients with HAE annually, this number is consistent with previous surveys of US physicians treating patients with HAE, in which the majority (70–74%) reported treating 1–5 patients with HAE in the past year [26]. The data presented are necessarily dependent on participant recall of events and self-reported perspectives, not actual practice behavior. The number of interviewees in the study (N = 15) was relatively small, yet the demonstration of saturation within our sample size is consistent with the following observations: (1) the group was highly homogeneous, (2) the data quality is high and consistent throughout, (3) the scope of inquiry was narrow, and (4) we did not attempt to demonstrate variation or correlation of data according to group variables. Results of our saturation analysis were consistent with existing literature on achieving saturation [14,16] and with our previous studies [12,13].

## Conclusion

This qualitative study revealed a range of barriers and promoters to research translation into clinical care for patients with HAE in the United States. The findings of this study have implications for the care of these patients. In addition to lowering institutional and practice-related barriers to research translation, there is an unmet need for patient and health professional education on HAE.

## Author contributions

**Conceptualization:** Esther L. Langmack, Dana Ravyn, Rob Lowney, William R. Lumry.

**Data curation:** Dana Ravyn, Beth Goodwin.

**Formal analysis:** Dana Ravyn.

**Funding acquisition:** Rob Lowney, Beth Goodwin.

**Investigation:** Esther L. Langmack, Dana Ravyn, William R. Lumry.

**Methodology:** Dana Ravyn.

**Project administration:** Rob Lowney, Beth Goodwin.

**Resources:** Rob Lowney, Beth Goodwin.

**Software:** Dana Ravyn.

**Supervision:** Rob Lowney, William R. Lumry.

**Writing – original draft:** Esther L. Langmack, Dana Ravyn, William R. Lumry.

**Writing – review & editing:** Esther L. Langmack, Dana Ravyn, Rob Lowney, Beth Goodwin, William R. Lumry.

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
