## [Decision Letter · Decision Letter 0]

9 May 2025

Dear Dr. Langmack,

Thank you for submitting your manuscript to PLOS One. Firstly, we would like to apologize for the delay in processing your manuscript. It has been exceptionally difficult to secure reviewers to evaluate your study. We have now received one completed review, which is available below. The reviewer has raised significant scientific concerns about the study that need to be addressed in a revision.

Please note that we have only been able to secure a single reviewer to assess your manuscript. We are issuing a decision on your manuscript at this point to prevent further delays in the evaluation of your manuscript. Please be aware that the editor who handles your revised manuscript might find it necessary to invite additional reviewers to assess this work once the revised manuscript is submitted. However, we will aim to proceed on the basis of this single review if possible. 

We look forward to receiving your revised manuscript.

Kind regards,

Miquel Vall-llosera Camps

Senior Staff Editor

PLOS ONE

Journal Requirements:

3. Thank you for stating the following in the Competing Interests section: [I have read the journal's policy and the authors of this manuscript have the following competing interests. Esther L. Langmack, Dana Ravyn, and Beth Goodwin state they have no competing interests to disclose.

Rob Lowney discloses that this research was supported by an independent educational grant from Takeda Pharmaceuticals USA, Inc. (Grant IME-007727) to CMEology, West Hartford, CT.

William R. Lumry discloses the following: grants or contracts (Astria, BioMarin, CSL Behring, Intellia, Grifols, Ionis, Kalvista, Shire/Takeda); consulting fees (Astria, BioCryst, Biomarin, CSL Behring, Express Scripts/CVS, Fresenius Kabi, Intellia, Kalvista, Magellan, Optum, Pharming, Pharvaris, Shire/Takeda); payments or honoraria for lectures, etc. (BioCryst, CSL Behring, Optinose, Pharming, Shire/Takeda, Grifols, Astra Zeneca, Sanofi/Regeneron, GSK); payment for expert testimony (Vedder-Price, Murphy & King); leadership for fiduciary roles (US Hereditary Angioedema Association Medical Advisory Board).]. 

Reviewers' comments:

Reviewer's Responses to Questions

**Comments to the Author**

1. Is the manuscript technically sound, and do the data support the conclusions?

Reviewer #1: Yes

2. Has the statistical analysis been performed appropriately and rigorously?

Reviewer #1: N/A

3. Have the authors made all data underlying the findings in their manuscript fully available?

Reviewer #1: Yes

4. Is the manuscript presented in an intelligible fashion and written in standard English?

Reviewer #1: Yes

Reviewer #1: In this manuscript by Langmark et al., significant issues related to the translation in clinical practice of research topics related to hereditary angioedema (HAE) management were analyzed by an interview approach with allergy/immunologists practicing in the USA. The scope of the work was to highlight barriers and promoters of such translations. Major barriers that emerged from the interviews were 1) insurance prior authorization for medications, including drugs for long-term prophylaxis of HAE attacks, 2) laboratory testing difficulties for HAE diagnosis, 3) time constraints for physicians and patient care; while promoters of translations were: 1) availability of medications samples, 2) route/frequency of administration (for LTP), and 3) shared-decision making.

The study, presented using an overall narrative approach, adds some relevant information on the clinical management of HAE patients. Notwithstanding, it applies to the specific USA healthcare setting, whereby, for example, the issue of insurance prior authorization for medications does not apply in Countries based on a public healthcare system.

Based on the mean experience of interviewed physicians, it is clear that they are not HAE specialists. HAE is a true rare disease and is handled in many Countries (especially in Europe) within a network of rare disease centers with HAE specialists who take care of patients.

I would therefore suggest to the AA. to recall in the title that the study was carried on in the USA.

Also, I would suggest making a comment in the discussion about the specific setting of this study, whereby general allergy specialists or general practitioners participated in the study, therefore, there could be a different scenario if the study had been carried out with HAE specialists.

**Do you want your identity to be public for this peer review?** For information about this choice, including consent withdrawal, please see our Privacy Policy

Reviewer #1: **Yes: ** Vincenzo Montinaro

---

## [Author Response · Author response to Decision Letter 1]

26 May 2025

RESPONSE TO REVIEWERS

Re: PONE-D-25-13647

May 24, 2025

To: Miquel Vall-Illosera Camps

Senior Staff Editor

PLOS One

Thank you for your kind consideration of our manuscript (PONE-D-25-13647).

Please be aware that this manuscript now has a new title: Barriers and promoters to adapting research findings to clinical care in hereditary angioedema in the U.S.: a qualitative study. The title was changed in response to a Reviewer #1 comment.

We appreciate the editors’ and the reviewer’s constructive suggestions and have made the following changes in response.

1. We modified the manuscript to meet PLOS One’s style requirements. A detailed record of all changes can be viewed in the uploaded Revised Manuscript with Track Changes.

2. We uploaded our de-identified, anonymized data to Harvard Dataverse, a repository recommended by PLOS One. The DOI for our dataset is: https://doi.org/10.7910/DVN/ZBIFEF

a. The dataset DOI is mentioned and cited in the Methods section of the paper (Manuscript, Line 109)

b. The reference list contains a citation for the dataset (Reference 9, Manuscript, Line 571)

c. The Data Availability statement in the submission form now reflects the new dataset repository

3. At the request of the Journal editors, we have added the statement “This does not alter our adherence to PLOS One policies on sharing data and materials” to the Competing Interests statement in our cover letter. Please see the Cover Letter submitted today for the revised Competing Interests statement.

4. We responded to the Reviewer Comments to the Author, as described below.

Comments to the Author

1. Is the manuscript technically sound, and do the data support the conclusions?

Reviewer #1: Yes

2. Has the statistical analysis been performed appropriately and rigorously?

Reviewer #1: N/A

3. Have the authors made all data underlying the findings in their manuscript fully available?

AUTHOR RESPONSE: We have uploaded our de-identified, anonymized data to Harvard Dataverse, a repository recommended by PLOS One. The DOI for our dataset is: https://doi.org/10.7910/DVN/ZBIFEF The dataset has been cited in the Methods section of the manuscript (Manuscript, Line 108-109), and a citation for the dataset repository appears in the reference list (Manuscript, Reference 9, Line 571).

4. Is the manuscript presented in an intelligible fashion and written in standard English?

Reviewer #1: Yes

5. Review Comments to the Author

Reviewer #1: In this manuscript by Langmark et al., significant issues related to the translation in clinical practice of research topics related to hereditary angioedema (HAE) management were analyzed by an interview approach with allergy/immunologists practicing in the USA. The scope of the work was to highlight barriers and promoters of such translations. Major barriers that emerged from the interviews were 1) insurance prior authorization for medications, including drugs for long-term prophylaxis of HAE attacks, 2) laboratory testing difficulties for HAE diagnosis, 3) time constraints for physicians and patient care; while promoters of translations were: 1) availability of medications samples, 2) route/frequency of administration (for LTP), and 3) shared-decision making.

The study, presented using an overall narrative approach, adds some relevant information on the clinical management of HAE patients. Notwithstanding, it applies to the specific USA healthcare setting, whereby, for example, the issue of insurance prior authorization for medications does not apply in Countries based on a public healthcare system.

Based on the mean experience of interviewed physicians, it is clear that they are not HAE specialists. HAE is a true rare disease and is handled in many Countries (especially in Europe) within a network of rare disease centers with HAE specialists who take care of patients.

I would therefore suggest to the AA. to recall in the title that the study was carried on in the USA.

Also, I would suggest making a comment in the discussion about the specific setting of this study, whereby general allergy specialists or general practitioners participated in the study, therefore, there could be a different scenario if the study had been carried out with HAE specialists.

AUTHOR RESPONSE: We agree that clinical practices and challenges associated with HAE care vary around the world and that it is necessary to stipulate that our study involved health care providers located only in the United States. Accordingly, we have made the following changes:

a. As suggested by Reviewer #1, we changed the title from “Barriers and promoters to adapting research findings to clinical care in hereditary angioedema: a qualitative study” to “Barriers and promoters to adapting research findings to clinical care in hereditary angioedema in the U.S.: a qualitative study” (Manuscript, Line 1)

b. As suggested by Reviewer #1, we specifically acknowledge that U.S. clinical practice characteristics and challenges may differ from those elsewhere in the world in the Discussion (Manuscript, Line 525-530), which has been changed to read as follows (changes are underlined):

Potential limitations to this study include the fact that it enrolled a non-random sample of US allergy/immunology health professionals, the majority of whom were general allergy practitioners in group practices, whose perspectives may potentially differ from those of a broader group of providers. Barriers and promoters relevant to research translation into HAE clinical practice may be different outside of the United States, for example in Europe, where many patients with HAE receive care in rare disease centers from HAE specialists rather than general allergy practitioners. Additionally, insurance prior authorization would not be a barrier in countries in which the public healthcare system does not require this type of review.

c. We amended the Conclusion to acknowledge that our research relates to HAE clinical care in the United States (Manuscript, Line 541-542), as follows (changes are underlined):

This qualitative study revealed a range of barriers to and promoters of research translation into clinical care for patients with HAE in the United States.

END of RESPONSE TO REVIEWER DOCUMENT

---

## [Decision Letter · Decision Letter 1]

5 Aug 2025

Dear Dr. Langmack,

Thank you for submitting your manuscript to PLOS ONE. After careful consideration, we feel that it has merit but does not fully meet PLOS ONE’s publication criteria as it currently stands. Therefore, we invite you to submit a revised version of the manuscript that addresses the points raised during the review process.

We look forward to receiving your revised manuscript.

Kind regards,

Miquel Vall-llosera Camps

Senior Staff Editor

PLOS ONE

Journal Requirements:

Additional Editor Comments:

Please address Reviewer#2 comments

Reviewers' comments:

Reviewer's Responses to Questions

**Comments to the Author**

Reviewer #1: All comments have been addressed

Reviewer #2: All comments have been addressed

2. Is the manuscript technically sound, and do the data support the conclusions?

Reviewer #1: Yes

Reviewer #2: Yes

3. Has the statistical analysis been performed appropriately and rigorously?

Reviewer #1: N/A

Reviewer #2: Yes

4. Have the authors made all data underlying the findings in their manuscript fully available?

Reviewer #1: Yes

Reviewer #2: Yes

5. Is the manuscript presented in an intelligible fashion and written in standard English?

Reviewer #1: Yes

Reviewer #2: Yes

Reviewer #1: The AA. have properly edited the manuscript and addressed the critiques raised to the first submitted version.

Reviewer #2: Dear authors, the novelty reported in your paper is limited.

As stated also by another reviewer, the experience of the interviewed physician was low; the majority reported following fewer than five patients. For this reason, the insights reported in the interview are all well known to all the physicians with a larger number of HAE patients.

The authors have pointed out in their paper that the field of HAE publication is quite saturated; therefore, I believe it may be worth considering whether this paper adds significant value to the existing body of work.

**Do you want your identity to be public for this peer review?** For information about this choice, including consent withdrawal, please see our Privacy Policy

Reviewer #1: **Yes: ** Vincenzo Montinaro

Reviewer #2: No

---

## [Author Response · Author response to Decision Letter 2]

25 Aug 2025

RESPONSE TO REVIEWERS

Re: PONE-D-25-13647

August 25, 2025

To: Miquel Vall-Illosera Camps

Senior Staff Editor

PLOS One

Thank you again for your kind consideration of our manuscript (PONE-D-25-13647).

We are grateful for both of the Reviewers’ constructive suggestions.

We have further revised the manuscript in response to Reviewer #2’s suggestions, as described below.

Comments to the Author

1. If the authors have adequately addressed your comments raised in a previous round of review and you feel that this manuscript is now acceptable for publication, you may indicate that here to bypass the “Comments to the Author” section, enter your conflict of interest statement in the “Confidential to Editor” section, and submit your "Accept" recommendation.

Reviewer #1: All comments have been addressed

Reviewer #2: All comments have been addressed

2. Is the manuscript technically sound, and do the data support the conclusions?

Reviewer #1: Yes

Reviewer #2: Yes

3. Has the statistical analysis been performed appropriately and rigorously?

Reviewer #1: N/A

Reviewer #2: Yes

4. Have the authors made all data underlying the findings in their manuscript fully available?

Reviewer #1: Yes

Reviewer #2: Yes

5. Is the manuscript presented in an intelligible fashion and written in standard English?

Reviewer #1: Yes

Reviewer #2: Yes

6. Review Comments to the Author

Reviewer #1: The AA. have properly edited the manuscript and addressed the critiques raised to the first submitted

Reviewer #2: Dear authors, the novelty reported in your paper is limited.

As stated also by another reviewer, the experience of the interviewed physician was low; the majority reported following fewer than five patients. For this reason, the insights reported in the interview are all well known to all the physicians with a larger number of HAE patients. The authors have pointed out in their paper that the field of HAE publication is quite saturated; therefore, I believe it may be worth considering whether this paper adds significant value to the existing body of work.

AUTHOR RESPONSE:

Little has been previously published about the clinician experience in HAE care. Most research has focused on the patient experience with HAE. In part because HAE is a rare disease, much less is known about clinician challenges in HAE care compared with a common condition like asthma.

We therefore do not consider this area of HAE research (clinician challenges) to be saturated. In fact, we would assert that clinician challenges with HAE need to be more deeply understood before medical care can be improved. Our study, which is novel in its application of qualitative research methods to the topic at hand, provides new information that builds on the findings of previous studies (lines 1031-1042, Track Changes version).

It is true that clinicians in our study population treated 5 patients with HAE on average annually. However, our data align closely with previous US surveys showing that the majority (70–74%) of US allergy/immunology clinicians treat only 1 to 5 patients with HAE annually. HAE is a rare disease (estimated prevalence of 1 in 50,000 person). “Physicians with a larger number of HAE patients” practice at HAE subspecialty/referral centers. They would also benefit from knowing more about the specific challenges faced by general allergy practice clinicians like those interviewed in our study.

To better explain the value of our findings — and how qualitative research can augment the body of knowledge on clinician perspectives — we have made the following changes. (References have been updated accordingly.)

a. We clarified how our study fits within the context of previous studies in the Discussion.

Lines 568-579, Track Changes version:

The results of this study build and expand upon the findings of quantitative survey studies of allergy/immunology health professionals [20, 21]. One study described challenges related to misdiagnosis of HAE, misinterpretation of test results, and other problems with laboratory testing [20]. Additional challenges were lack of clinician and patient knowledge of HAE, patient inability to afford or access treatment, and difficulties incorporating patient preferences into treatment decisions [20]. Regarding long-term HAE prophylaxis, physicians and patients were previously shown to have differing views of treatment burden [21]. Physicians were more likely than patients themselves to view prophylactic medications administered subcutaneously or intravenously as burdensome, inconvenient, or unpleasant. These findings and those of our study highlight the need for enhanced patient-physician communication and shared decision-making about long-term prophylaxis options.

b. We articulated more clearly the strengths of our study in a new paragraph in the Discussion.

Lines 609-630, Track Changes version:

One strength of this study is that we uncovered several factors not previously reported in quantitative survey studies of allergy/immunology clinicians that may impact delivery of high-quality HAE patient care [20, 21]. Specifically, we detected clinician difficulties with accessing information on HAE, the use of online chat rooms as an information resource, unfamiliarity with HAE clinical practice guidelines, and lack of use of QoL assessment tools as barriers to research translation. Provision of medication samples and pharmaceutical representative detailing were identified as facilitators of care delivery in our study but not previous quantitative survey studies [20, 21]. In translational health research, qualitative methods are increasingly recognized as valuable ways to reveal stakeholder attitudes, perceptions, and experiences that escape detection using quantitative approaches [23]. Additionally, this study is the first, to our knowledge, to use contemporary qualitative research methods including computer-assisted thematic analysis to examine the experiences of allergy/immunology clinicians related to HAE patient care. Recent qualitative research on HAE has focused on patients rather than clinicians [3, 22, 24,25]. However, the experiences and perspectives of allergy/immunology clinicians, particularly those in general allergy/immunology practices (where most of our respondents worked), also need to be more fully recognized if treatment is to improve for all HAE patients, not just those who receive care in HAE subspecialty centers.

c. We added this statement about the number of patients with HAE treated annually to the Discussion.

Lines 640-643, Track Changes version:

Although clinicians in our study reported seeing an average of only 5 patients with HAE annually, this number is consistent with previous surveys of US physicians treating patients with HAE, in which the majority (70–74%) reported treating 1 to 5 patients with HAE in the past year [26].

7. Do you want your identity to be public for this peer review?

Reviewer #1: Yes: Vincenzo Montinaro

Reviewer #2: No

END of RESPONSE TO REVIEWER DOCUMENT

---

## [Decision Letter · Decision Letter 2]

23 Sep 2025

Barriers and promoters to adapting research findings to clinical care in hereditary angioedema in the U.S.: a qualitative study

PONE-D-25-13647R2

Dear Dr. Langmack,

We’re pleased to inform you that your manuscript has been judged scientifically suitable for publication and will be formally accepted for publication once it meets all outstanding technical requirements.

Kind regards,

Miquel Vall-llosera Camps

Staff Editor

PLOS One

Additional Editor Comments:

We consider that the previous concerns have been sufficiently addressed.

Reviewers' comments:

Reviewer's Responses to Questions

**Comments to the Author**

Reviewer #1: (No Response)

Reviewer #2: (No Response)

2. Is the manuscript technically sound, and do the data support the conclusions?

Reviewer #1: Yes

Reviewer #2: No

3. Has the statistical analysis been performed appropriately and rigorously?

Reviewer #1: N/A

Reviewer #2: Yes

4. Have the authors made all data underlying the findings in their manuscript fully available?

Reviewer #1: Yes

Reviewer #2: Yes

5. Is the manuscript presented in an intelligible fashion and written in standard English?

Reviewer #1: Yes

Reviewer #2: Yes

Reviewer #1: the manuscript has significantly improved in its present form. The AA. have answered to all comments made to the original version

Reviewer #2: Dear Authors, the number of papers published about HAE is increasing, and most of these papers have not significantly contributed to HAE knowledge. If you search on PubMed:

https://pubmed.ncbi.nlm.nih.gov/?term=hereditary+angioedema&sort=date

In the first eight months of the year, 181 papers were published.

**Do you want your identity to be public for this peer review?** For information about this choice, including consent withdrawal, please see our Privacy Policy

Reviewer #1: **Yes: ** Vincenzo Montinaro

Reviewer #2: No

---

## [Editor Report · Acceptance letter]

PONE-D-25-13647R2

PLOS ONE

Dear Dr. Langmack,

I'm pleased to inform you that your manuscript has been deemed suitable for publication in PLOS ONE. Congratulations! Your manuscript is now being handed over to our production team.

Kind regards,

on behalf of

Dr. PLOS Manuscript Reassignment

Staff Editor

PLOS ONE